A miRNome analysis at the early postmortem interval

http://orcid.org/0000-0001-8790-5504 Guardado-Estrada Mariano 1 mguardado@cienciaforense.facmed.unam.mx
Cárdenas-Monroy Christian A. 1
Martínez-Rivera Vanessa 1
Cortez Fernanda 2
Pedraza-Lara Carlos 3
Millan-Catalan Oliver 4
Pérez-Plasencia Carlos 4 5
1 Laboratorio de Genética, Ciencia Forense, Facultad de Medicina, Universidad Nacional Autónoma de México , Mexico City , Mexico
2 Computational Genomics Division, Instituto Nacional de Medicina Genómica (INMEGEN) , Mexico City , Mexico
3 Laboratorio de Entomología, Ciencia Forense, Facultad de Medicina, Universidad Nacional Autónoma de México , Mexico City , Mexico
4 Unidad de Investigación Biomédica en Cáncer, Laboratorio de Genómica, Instituto Nacional de Cancerología , Mexico City , Mexico
5 Unidad de Investigación Biomédica en Cáncer, Laboratorio de Genómica, Facultad de Estudios Superiores Iztacala, Universidad Nacional Autónoma de México , Mexico City , Mexico
Davalos Alberto
Electronic publication date: 2023 Jun 7
Publication date: 2023
Volume: 11
Electronic Location ID: e15409
Received 2023 Jan 24; Accepted 2023 Apr 23
Copyright: © 2023 Guardado-Estrada et al.
Copyright year: 2023
Copyright holder: Guardado-Estrada et al.
License: This is an open access article distributed under the terms of the Creative Commons Attribution License, which permits unrestricted use, distribution, reproduction and adaptation in any medium and for any purpose provided that it is properly attributed. For attribution, the original author(s), title, publication source (PeerJ) and either DOI or URL of the article must be cited.
License URL: https://creativecommons.org/licenses/by/4.0/

Keywords: Postmortem interval, miRNA, Microarrays, Non-coding RNA

Funding: National Autonomous University of Mexico (UNAM), PAPIIT IA204420 This work was supported by the National Autonomous University of Mexico (UNAM), PAPIIT grant number IA204420. The funders had no role in study design, data collection and analysis, decision to publish, or preparation of the manuscript.

==============================
The postmortem interval (PMI) is the time elapsing since the death of an individual until the body is examined. Different molecules have been analyzed to better estimate the PMI with variable results. The miRNAs draw attention in the forensic field to estimate the PMI as they can better support degradation. In the present work, we analyzed the miRNome at early PMI in rats’ skeletal muscle using the Affymetrix GeneChip™ miRNA 4.0 microarrays. We found 156 dysregulated miRNAs in rats’ skeletal muscle at 24 h of PMI, out of which 84 were downregulated, and 72 upregulated. The miRNA most significantly downregulated was miR-139-5p (FC = −160, p = 9.97 × 10−11), while the most upregulated was rno-miR-92b-5p (FC = 241.18, p = 2.39 × 10−6). Regarding the targets of these dysregulated miRNAs, the rno-miR-125b-5p and rno-miR-138-5p were the miRNAs with more mRNA targets. The mRNA targets that we found in the present study participate in several biological processes such as interleukin secretion regulation, translation regulation, cell growth, or low oxygen response. In addition, we found a downregulation of SIRT1 mRNA and an upregulation of TGFBR2 mRNA at 24 h of PMI. These results suggest there is an active participation of miRNAs at early PMI which could be further explored to identify potential biomarkers for PMI estimation.

Introduction

The postmortem interval (PMI) is the time elapsing since the death of an individual until its body is analyzed (Mathur & Agrawal, 2011). The calculation of this time is relevant in the forensic field due to its estimation could be helpful in solving criminal cases. Although the identification of several physical parameters has been used to estimate the PMI (temperature, coloration, stiffness), the occurrence of these physical changes in the dead body depends on internal and external factors (Powers, 2005; Mathur & Agrawal, 2011). Thus, other molecular biomarkers, including RNAs, have been analyzed to estimate the PMI with variable results (Madea, Kreuser & Banaschak, 2001; Sampaio-Silva et al., 2013; Donaldson & Lamont, 2013; Ansari & Menon, 2017).

It is well known that it is possible to find some mRNAs at the postmortem interval (Li et al., 2014; Lv et al., 2014; Nagy et al., 2015; Tu et al., 2018, 2019; Wang et al., 2019). In fact, changes in the mRNA expression of several genes at the PMI have been reported in different species, including humans (Pozhitkov et al., 2017; Ferreira et al., 2018). The postmortem expression of these mRNAs seems to be related to the cellular processes that occur in the body decomposition such as apoptosis, autolysis, or immune response (Zapico, Menéndez & Núñez, 2014; Haas et al., 2021; Dachet et al., 2021). The miRNAs are non-coding RNAs with a length between 20–22 nucleotides that regulate the expression of mRNAs (Saliminejad et al., 2019). Due to their physical characteristics and their small size, miRNAs can better support degradation making its analysis suitable for forensic purposes (Rocchi et al., 2021). For instance, miRNAs have been used to both identify body fluids and in postmortem studies (Wang et al., 2013b). Regarding the postmortem studies, a continuous expression of some miRNAs at different PMI in different tissues has been found both in rats and humans (Maiese et al., 2021). These studies have been performed to find those miRNAs that could be useful as housekeeping genes or to correlate their degradation with the PMI. Although it is expected that the expression of some of these miRNAs decays as the PMI increases, there are miRNAs whose expression increases at PMI, such as MiR195 and MiR122 (Wang et al., 2013a). In fact, in our lab we found an increase in the expression of miR-381-3p in rats’ skeletal muscle at the 24 h of PMI (Martínez-Rivera et al., 2021). This suggests that just as with some mRNAs, there is an upregulation of miRNAs which are needed for the body decomposition process. Moreover, it is probable that other miRNAs participate in this process, and that their identification could be important to identify possible biomarkers or to normalize genes for PMI gene expression studies. To identify the miRNAs that participate in the PMI, we analyzed the miRNome using rat microarrays in rats’ skeletal muscle at 24 h of PMI. Also, we performed an in-silico analysis to identify the targets of these miRNAs and the biological process where they participate. Finally, we analyzed the gene expression of three miRNAs targets by quantitative RT-PCR.

Materials and Methods

Samples

A total of five rat skeletal muscle samples were obtained from a postmortem interval rat model which was previously established in our laboratory (Martínez-Rivera et al., 2021). In brief, nine adult male Wistar rats were selected for the study; all with an average weight of 200 gr. These rats were obtained from the Faculty of Medicine’s animal facility of the National Autonomous University of Mexico (UNAM). Previously to the study, these rats were maintained in a cage with no food or water restrictions, having a 12-h light-dark cycle and controlled temperature (22 °C) and humidity (40–60%). These rats were divided into two groups which corresponded to 0 (n = 4) and 24 h of PMI (n = 5). The 0 h-PMI was considered the control group. The procedures to obtain the muscle samples of rats at 24 h of PMI and controls were performed as previously described (Martínez-Rivera et al., 2021). All procedures of the PMI rat model reported in this study were approved by the local ethic and scientific committees (Martínez-Rivera et al., 2021).

RNA extraction

The RNAs from each rat skeletal muscle sample were obtained using glass beads and Trizol™ Reagent as previously described (Martínez-Rivera et al., 2021). The integrity of the obtained RNAs was evaluated in agarose gels (Fig. S1) (Martínez-Rivera et al., 2021).

The miRNome analysis

To fully analyze the miRNome in the skeletal muscle samples, we used the Affymetrix GeneChip miRNA 4.0 Array (Affymetrix, Santa Clara, CA, USA). The Affymetrix GeneChip miRNA 4.0 Array contains a total of 30,424 probes of mature miRNAs, out of which 1,218 correspond to rat miRNAs. All experiments were in accordance with the manufacturer’s protocol, and these experiments were performed at the Microarray Unit of the Mexican National Institute of Genomic Medicine (INMEGEN). In brief, the obtained RNAs from skeletal muscle were biotin labeled with the Affymetrix® FlashTag™ Biotin HSR RNA Labeling Kit according to the manufacturer protocol. Labeled samples were hybridized with Affymetrix GeneChip miRNA 4.0 Array (Affymetrix, Santa Clara, CA, USA). The hybridized microarrays were washed and stained with the Affymetrix GeneChip hybridization wash and stain kit, and then scanned with the Affymetrix GeneChip Scanner 3000 7G to generate de CEL files.

The CEL files were analyzed with the Affymetrix Transcriptome Analysis Console (TAC) Software™. With this software, the probes intensities of the miRNAs were normalized, and the miRNA gene expression was calculated with the Robust Multi-chip Analysis (RMA) setting the value of Detected Above Background (DABG) to 0.05. The changes of gene expression were shown as mean Fold Change (FC), considering miRNAs over or down expressed with thresholds above >2 or below <2, respectively. Significant dysregulated miRNAs were those with p-value less than 0.01 and a False Discovery Rate value less than 0.05.

miRNAs target analysis and their biological processes

The targets of the significant dysregulated miRNAs were explored with the webtool MIcroRNA Enrichment Turned Network (http://userver.bio.uniroma1.it/apps/mienturnet/) (Licursi et al., 2019). This tool identifies computationally predicted or experimentally validated miRNA-mRNA interactions downloaded from TargetScan and miRTarBAse (McGeary et al., 2019; Huang et al., 2020). All parameters were set as default. The miRNAs with most mRNA interactions were visualized using the Cytoscape software (Shannon et al., 2003). Moreover, to identify the biological pathways where these mRNA targets participate, we further analyzed them with the Gene Set Enrichment Analysis (GSEA) from WEB-based Gene Set Analysis Toolkit (WebGestalt, http://www.webgestalt.org/).

mRNA quantification by RT-PCR

The cDNA was synthesized from the RNAs extracted from the rats’ muscle samples as previously described (González Ramírez et al., 2020). A total of 45 ng of each cDNA synthesized was employed to relatively quantify the gene expression of TGFBR2, SIRT1 and BMF with RT-PCR using SYBR™ Green PowerUp™ (Thermo Fisher Scientific, Waltham, MA, USA) according to the manufacturer’s protocol. To normalize the expression of these genes, the GAPDH gene was used in the analysis as a housekeeping gene. Each gene was analyzed separately and ran by triplicate in all samples. The gene expression of these genes was calculated with the average CT threshold of each sample and expressed as Fold-Change (FC) (Livak & Schmittgen, 2001). The sequence of the oligos used to analyze the expression of TGFBR2, SIRT1, BMF and GAPDH are in Table S5.

Results

Muscle rat miRNome analysis at 24 h of postmortem interval

The miRNome rats’ skeletal muscle was analyzed at 24 h of postmortem interval using the Affymetrix GeneChip™ miRNA 4.0 Array and compared with a control group. From the 1,218 analyzed miRNAs, we found 84 (6.7%) downregulated and 72 (5.8%) upregulated miRNAs in skeletal rat muscle after the first 24 h of the postmortem interval (see Fig. 1 and Table S1). It is noteworthy that although the frequency of downregulated miRNAs was slightly higher than the upregulated, this difference was not significant (p > 0.05, chi-square test). With respect to the miRNome gene expression, the average fold change of downregulated miRNas was −9.2, while in the upregulated miRNas was 20.3. The top ten most significant upregulated and downregulated miRNAs are in Table 1. From these dysregulated miRNAs, the most significant upregulated miRNAs were rno-miR-92b-5p (FC = 241.2, p = 2.39 × 10−6), rno-miR-297 (187.2, p = 2.25 × 10−9), rno-miR-466b-5p (FC = 122.4, p = 8.98 × 10−8), rno-miR-466c-5p (FC = 121.2, p = 2.06 × 10−8), and rno-miR-466d (FC = 92.8, p = 2.81 × 10−7; see Table 1). Meanwhile, the most downregulated miRNA after the 24 h of postmortem interval was rno-miR-139-5p (FC = −160, p = 9.97 × 10−11).

Figure 1 Volcano plot of miRNAs expression at 24 h of PMI.

In this graph the p-value (−log10) is plotted against the fold change (FC) of the 1,218 rat miRNAs included in the microarray. Red dots indicate upregulated miRNAs, while blue dots the downregulated. The hightlighted dots indicate the top most significant up and down regulated miRNAs.

Table 1 Top ten most up and down-regulated miRNAs at 24 h of post-mortem interval.

miRNA	Fold change
(FC)	Affymetrix
probe set ID	p-value	FDR p-value	
Up-regulated					
rno-miR-92b-5p	241.18	20506567	2.4 × 10−6	8 × 10−5	
rno-miR-297	187.25	20501565	2.2 × 10−9	5.9 × 10−7	
rno-miR-466b-5p /// rno-miR-466b-5p	122.4	20506476	9 × 10−8	5.1 × 10−6	
rno-miR-466c-5p	121.18	20506479	2 × 10−8	1.8 × 10−6	
rno-miR-466d	92.77	20517088	2.8 × 10−7	1.4 × 10−5	
rno-miR-465-5p	59.38	20513753	6 × 10−9	1 × 10−6	
rno-miR-32-3p	55.86	20501395	5.9 × 10−8	3.9 × 10−6	
rno-miR-702-5p	34.08	20517148	1.6 × 10−5	0.0003	
rno-miR-195-3p	33.44	20501513	6.8 × 10−9	1 × 10−6	
rno-miR-326-5p	32.97	20500944	8.9 × 10−9	1. × 10−6	
Down-regulated					
rno-miR-139-5p	−160.08	20501464	10 × 10−11	6.2 × 10−8	
rno-miR-532-3p	−36.3	20506543	6.9 × 10−5	0.0007	
rno-miR-342-3p	−33.74	20500994	0.0001	0.001	
rno-miR-433-3p	−29.16	20502443	4.3 × 10−5	0.0005	
rno-miR-328a-3p	−23.1	20500950	0.002	0.008	
rno-miR-339-5p	−17.44	20500982	0.00004	0.0006	
rno-miR-455-3p	−16.11	20506533	0.000015	0.0003	
rno-miR-222-3p	−15.57	20501555	0.0001	0.001	
rno-miR-431	−15.38	20502439	0.0006	0.003	
rno-mir-423	−14.41	20535731	2 × 10−6	7.5 × 10−5	

Interestingly, the gene expression of the most significant miRNAs could differentiate the 24 h PMI group from the control group as seen in the dendrogram (see Fig. 2). In this dendrogram, we could see a cluster of miRNAs where the expression was up-regulated at 24 h-PMI in comparison with the 0 h-PMI, while there were also groups of miRNAs which were downregulated at 24 h-PMI. This was also corroborated in a PCA plot analyzing the whole miRNA gene expression (Fig. S2). These data indicate that at 24 h of postmortem interval there are changes in the gene expression of several miRNAs in the rats’ skeletal muscle.

Figure 2 Hierarchical clustering of the 156 differentially expressed miRNAs.

The fold changes of the 156 differentially expressed miRNAs found at 24 h of PMI compared with controls were hierarchically clustered and represented as a dendrogram. Red color represents upregulated miRNAs and blue color downregulated miRNAs, and its intensity is related with the grade of miRNA expression.

Targets of differentially expressed miRNAs

The targets of the 156 differentially expressed miRNAs at 24 h of postmortem interval were identified using the MIENTURNET Bioinformatics Tool and visualized using Cytoscape. Interestingly, of the down-regulated miRNAs, there were only 16 miRNAs which interacted with at least two mRNAs (see Fig. 3). The miRNAs that had the most mRNA interactions were rno-miR-125b-5p/rno-miR-351-5p, and rno-miR-138-5p with 130 and 120 mRNAs, respectively (Table S2). The rest of the miRNAs with more than 20 interactions were rno-miR-92a-3p, rno-miR-133a-3p, rno-miR-133b-3p, rno-miR-93-5p, rno-miR-22-3p, rno-miR-26b-5p, rno-miR-181c-5p, rno-miR-181d-5p, rno-miR-221-3p, rno-miR-222-3p, and rno-miR-140-5p. These groups of miRNAs were connected among them by at least one mRNA target. The miRNAs that shared most mRNAs were rno-miR-22-3p, rno-miR-26b-5p, rno-miR-92a-3p, and rno-miR-93-5p (see Fig. 3). Although there were 72 miRNAs significantly upregulated at 24 h of PMI, only the rno-miR-291a-3p had more than one interaction with other mRNAs (n = 44; Table S2).

Figure 3 Network of the mRNA targets of downregulated miRNAs found at 24 h of PMI.

The targets of the miRNAs rno-miR-125b-5p, rno-miR-138-5p, rno-miR-92a-3p, rno-miR-133a-3p, rno-miR-133b-3p, rno-miR-93-5p, rno-miR-22-3p, rno-miR-26b-5p, rno-miR-181c-5p, rno-miR-181d-5p, rno-miR-221-3p, rno-miR-222-3p, rno-miR-140-5p, rno-miR-191a-5p, and rno-miR-455-3p are shown. The green circles represent the miRNA, and blue squares their mRNA target.

Gene ontology analysis

The gene targets of the dysregulated miRNAs that presented most mRNA interactions were analyzed with the WEB-based Gene Set Analysis Tool kit to identify the biological process where they participate. The main biological processes where the miRNAs rno-miR-125b-5p/rno-miR-351-5p, rno-miR-138-5p, rno-miR-92a-3p, rno-miR-133a-3p/rno-miR-133b-3p, rno-miR-93-5p, rno-miR-22-3p, rno-miR-26b-5p, rno-miR-181c-5p/rno-miR-181d-5p, rno-miR-221-3p/rno-miR-222-3p, rno-miR-140-5p, rno-miR-191a-5p, and rno-miR-455-3p participate, included positive regulation of interleukin-2 production, nucleotide transmembrane transport, negative regulation of cytoplasmic translation, positive regulation of synaptic vesicle fusion to presynaptic active zone membrane, interleukin-10 secretion, negative regulation of cell growth involved in cardiac muscle cell development, positive regulation of fat cell differentiation, megakaryocyte development, and vascular associated smooth muscle cell migration, respectively (see Table S3). On the contrary, the main biological process associated with rno-miR-291a-3p was epithelium development. Although there were also biological processes which had lower enrichment ratio, we found some biological processes that could be related with the postmortem interval. For instance, rno-miR-22-3p and rno-miR-181c-5p/rno-miR-181d-5p also participate in biological processes related to hypoxia response, and decreased oxygen levels (see Fig. 4 and Table S3).

Figure 4 Gene ontology enrichment analysis.

The main biological pathways where the target genes of rno-miR-125b-5p/rno-miR-351-5p, rno-miR-92a-3p, rno-miR-22-3p, and rno-miR-291a-3p participate are shown. The x-axis corresponds to the enrichment ratio.

Gene expression of miRNAs targets

We analyzed by quantitative RT-PCR the gene expression of mRNAs TGFBR2, SIRT1, and BMF at 24 h of postmortem interval which are targets of rno-miR-291a-3p, rno-miR-125b-5p/rno-miR-351-5p, and rno-miR-138-5p, respectively (see material and methods). Regarding to TGFBR2, the average FC was 1.8 (s.d. 0.9) times higher at the 24 h of PMI in comparison with controls (FC = 1, s.d. 0.1), and this difference was statistically significant (p = 0.03175, Wilcoxon rank sum test, see Fig. 5). Although the gene expression of BMF was also higher at the 24 h of PMI when compared with controls, this difference was not statistically significant (FC = 5.2, s.d. 3 vs FC = 1.4, s.d. 0.8; p = 0.1111, Wilcoxon rank sum test). On the other hand, there was a significant reduction of SIRT1 gene expression at 24 h of PMI (FC = 0.23, s.d. 0.07 vs FC = 1.1, s.d. 0.6; p = 0.01587, Wilcoxon rank sum test, see Fig. 5).

Figure 5 Gene expression analysis of TGFBR2, SIRT1 and BMF genes in rats’ skeletal muscle at early post-mortem interval.

The fold change (FC) of these mRNA was analyzed in rats’ skeletal muscle at 24 h of PMI relative to the 0 h-PMI group, using quantitative RT-qPCR. The fold change was calculated with the 2−∆∆CT method using GAPDH as internal control. The black squares represent the mean FC from each group; the whisker corresponds to the 95% confidence interval, and the dots are the jittered FC of each sample. Comparisons between the PMI were done with the Wilcoxon rank sum test.

Discussion

This is the first work which analyzes the whole miRNome in rat skeletal muscle at the 24 h of postmortem interval (PMI). In the present work we found a total 156 dysregulated miRNAs in rat muscle at 24 h of PMI compared with controls. From these dysregulated miRNAs, 84 were downregulated and 72 upregulated. The gene expression of these dysregulated miRNAs could clearly differentiate the muscle samples at 24 h of PMI from controls. The most downregulated miRNA was rno-miR-139-5p, while the most upregulated miRNAs were rno-miR-92b-5p, rno-miR-297, rno-miR-466b-5p, rno-miR-466c-5p, and rno-miR-466d. Nevertheless, from the downregulated miRNAs rno-miR-125b-5p/rno-miR-351-5p and rno-miR-138-5p were the miRNAs with more mRNA targets. On the other hand, rno-miR-291a-3p was the only upregulated miRNA which interacted with several mRNA targets. The targets of the miRNAs with more mRNA interactions participate in several biological processes that involve interleukin secretion, translation regulation, cell growth regulation, or low oxygen response.

The miRNAs gene expression seems to play a crucial role at the early postmortem interval since some of them have been found to be dysregulated in this process. It was naturally believed that miRNAs could be found after long periods of time in the postmortem interval supporting degradation due to their physical characteristics (Lv et al., 2017; Zhao et al., 2021). For that reason, some miRNAs have been used as internal controls in gene expression analysis at postmortem interval studies (Tu et al., 2019). Nevertheless, it has been reported that the expression of some miRNAs changes throughout the postmortem interval (Wang et al., 2013a; Nagy et al., 2015; Na, 2020; Montanari et al., 2021; Martínez-Rivera et al., 2021). For instance, we have found a dysregulation in gene expression of miR-381-3p and miR-23b-3p in rat skeletal muscle during the first 24 h of PMI (Martínez-Rivera et al., 2021). In the present study, we found dysregulation in the expression of genes miR-195, miR-134, miR-133, miR-16, and miR-150, which have also been found dysregulated at the early PMI in bone, brain, and spleen in both rats and humans (Wang et al., 2013a; Nagy et al., 2015; Na, 2020). According to these studies and ours, the presence of miRNAs at the PMI seems to be related with the postmortem process.

On the other hand, the results of the present study could contribute to better choose those miRNAs selected as internal controls for gene expression analyses in postmortem interval studies in rats. For instance, several miRNAs have been used as control genes for normalization in the gene expression analysis, such as mir191 and mir-133a (Kakimoto et al., 2015; Tu et al., 2018). Nevertheless, we found a downregulation of the gene expression in these miRNAs at 24 h of PMI. In the case of mir-133a, this miRNA has also been reported as a part of the family of myomiR which are responsible of the plasticity and homeostasis of the skeletal muscle (McCarthy, 2011; Koutalianos et al., 2021). Thus, their use as internal controls in rat skeletal muscle PMI gene expression studies could not be suitable. Moreover, it is plausible that not all the reported miRNAs used as internal controls for PMI gene expression studies could be used in all tissues since their expression could vary according to the explored tissue.

It is noteworthy that from the dysregulated miRNAs found in the present study, there were few miRNAs which had the highest fold-change value (rno-miR-92b-5p, rno-miR-297, rno-miR-466b-5p, rno-miR-466c-5p and rno-miR-466d), which contrasted with only one miRNA (rno-miR-139-5p) that presented the lowest fold-change value of the downregulated miRNAs. Nevertheless, none of these miRNAs have been reported or associated with the postmortem interval. For instance, miR-466b-5p has been implicated in the pathogenesis of white matter injury resulting from a hypoxia ischemia insult in rats (Schober et al., 2015). On the other hand, miR-92b-5p and rno-miR-466c-5p have been associated with acute heart failure, and response to lipemia (Wu et al., 2018; Mantilla-Escalante et al., 2019). Interestingly, the miR-139-5p has been found associated with necroptosis in patients with hepatocellular carcinoma (Meng et al., 2022).

One approach that we made in the present study was to find those biologically significant miRNAs which had the most interactions with mRNAs. From these miRNAs, we found that the downregulated miRNAs rno-miR-125b-5p/rno-miR-351-5p, and rno-miR-138-5p; and the upregulated miRNA rno-miR-291a-3p were the miRNAs with most mRNA interactions. Although none of these miRNAs have been reported at the postmortem interval, their functions could be related with this process. In the case of miR-125b-5p, it has been found that this miRNA could protect neuronal injury after cerebral ischemia reperfusion in stroke (Chen et al., 2020). Moreover, a down-regulation of miR-125b-5p has also been reported in rat muscle cells with atrophy and denervation (Qiu et al., 2019). This contrasts with the reports on cancer, where downregulation of this miRNA has been found in several tumors such as hepatocellular carcinoma or bladder cancer, which is related to cell proliferation, migration, and invasion (Hua et al., 2019; Liu, Chen & Wang, 2020). The targets of rno-miR-125b-5p participate in the production of interleukin 2, which stimulates the immune response (Peerlings, Mimpen & Damoiseaux, 2021). Interestingly, it has been reported an increase in the gene expression activity of immune cells of postmortem brains (Dachet et al., 2021). According to TargetScan, miR-125b-5p had the same targets as rno-miR-351-5p, which is involved in skeletal muscle development along with other miRNAs such as miR-1a-3p, miR-133a-3p, miR-133b-3p, miR-206-3p, and miR-128-3p (Xie et al., 2018).

The expression of miR-138-5p seems to play an important role in autophagy due to the fact that its downregulation is necessary for the induction of cancer (Tian et al., 2017; Zhou et al., 2021). The process of autophagy is a survival mechanism which has been described under normal conditions and in different pathologies. This process allows the cells to survive entering in a “self-digestion” process which provides energy during nutrient deprivation and metabolic stress (Chang, 2020). It is noteworthy that we found a downregulation at the 24 h PMI of SIRT1, which is the target of the miRNAs rno-miR-138-5p and rno-miR-22-3p that were found downregulated in our study. It has been reported that in senescent cells there is a downregulation of SIRT1 protein due to autophagy throughout LC3 (Xu et al., 2020). In fact, at the postmortem interval, an over-expression of some genes associated with autophagy as LC3, ATG7 or ATG12 has been found (Martínez et al., 2019). Also, SIRT1 regulates pro-inflamatory response through modulation of NF-κB signaling (Kim, Silwal & Jo, 2022). Although downregulation of SIRT1 is important in the autolysis process at postmortem interval, its post-transcriptional regulation could be due to other mechanisms rather than microRNAs (Cao et al., 2020).

Although rno-miR-291a-3p, was the only up-regulated miRNA which had more interactions with mRNAs, there is no information about its participation at the PMI. The rno-miR-291a-3p is included in the miR-290-295 cluster and represents more than the 60% of miRNAs present in mouse embryonic stem cells (Yuan et al., 2017). Interestingly, this cluster of miRNAs regulates the metabolism of these stem cells sustaining an increase of glycolysis to keep their pluripotency (Varum et al., 2011; Yuan et al., 2017). When we analyzed the gene expression of TGFBR2, which is the target of rno-miR-291a-3p, contrary to what was expected, the expression of this mRNA was upregulated. However, it has been reported that TGFBR2 could be regulated by other miRNAs such as rno-miR-19a-3p or hsa-miR-93-5p which are located within the miR-17-92a cluster in chromosome 13 (Mogilyansky & Rigoutsos, 2013; Zou et al., 2016; Cai et al., 2021). In fact, we found a downregulation in the expression of miR-93-5p. This suggests that the expression of TGFBR2 at the postmortem interval could be regulated by the miR-17-92a cluster. The TGFBR2 gene has a dual activity where it can either be an oncogene favoring the epithelial to mesenchymal transition, or in some situations lead to apoptosis and cell cycle arrest (Hata & Chen, 2016; Lo Sardo et al., 2021). It is plausible that the latter occurs at the postmortem interval.

It is important to mention that the present results were performed in an animal model under controlled conditions that could be extremely different from a forensic scenario. On the other hand, in the present work we only considered the 24 h of PMI. Hence, it is important to include other PMIs before and after the 24 h to better capture the biological process regulated by miRNAs in the PMI. Besides, although we selected the skeletal muscle tissue for this analysis, it is also possible that the miRNAs expression could differ from other tissues. Thus, future studies should be performed in other tissues and under conditions that resemble real forensic scenarios. The knowledge of the molecular process that occurs at the PMI could help to better identify possible biomarkers to precisely estimate the PMI. Also, this will be useful to design kits suitable to rutinary analyze these biomarkers in the forensic field. Moreover, the contribution of these studies could be extrapolated in other biomedical fields like cancer, since there is the possibility of there being cells which use the same survival mechanisms.

Conclusions

There are some miRNAs that were found dysregulated at the 24 h of postmortem interval, being the downregulated slightly more frequent than the upregulated. From these dysregulated miRNAs, rno-miR-125b-5p, rno-miR-138-5p and rno-miR-291a-3p proved to be the miRNAs with most mRNA interactions. The ontology analysis and the analysis of the expression of SIRT1 and TGFBR2 genes suggests that the main pathways that participate at the postmortem interval are related with autophagy, cell cycle regulation, low oxygen response, among others. Our results propose that autophagy is a mechanism could occur at the early postmortem. This knowledge could be important to understand the biology of death and to identify candidate biomarkers to estimate postmortem interval.

Supplemental Information

Supplemental Information 1 Arrive file.

Click here for additional data file.

Supplemental Information 2 Sample signals of all analyzed miRNAs between controls and 24 h of postmortem interval samples.

Click here for additional data file.

Supplemental Information 3 Raw data of RT-PCR.

Click here for additional data file.

Supplemental Information 4 Principal components analysis of miRNA expression between controls and 24 h of postmortem interval.

Click here for additional data file.

Supplemental Information 5 Principal Component Analysis of the whole miRNAs analyzed at 24 h of PMI.

Click here for additional data file.

Supplemental Information 6 Fold Change of the most significant miRNAs found between controls and 24 h of postmortem interval.

Click here for additional data file.

Supplemental Information 7 miRNAs with more mRNA targets found at 24 h of postmortem interval.

Click here for additional data file.

Supplemental Information 8 Gene ontology analysis of the targets from miRNAs with more interactions at 24 h of postmortem interval.

Click here for additional data file.

Supplemental Information 9 The sequence of the oligos used to analyze the expression of TGFBR2, SIRT1, BMF and GAPDH.

The sequence of the oligos used to analyze the expression of TGFBR2, SIRT1, BMF and GAPDH using RT-PCR

Click here for additional data file.

Supplemental Information 10 MIAME.

Click here for additional data file.

Additional Information and Declarations

Competing Interests

Author Contributions

Animal Ethics

Microarray Data Deposition

Data Availability

The authors declare that they have no competing interests.

Mariano Guardado-Estrada conceived and designed the experiments, analyzed the data, prepared figures and/or tables, authored or reviewed drafts of the article, and approved the final draft.

Christian A. Cárdenas-Monroy performed the experiments, prepared figures and/or tables, and approved the final draft.

Vanessa Martínez-Rivera performed the experiments, analyzed the data, prepared figures and/or tables, and approved the final draft.

Fernanda Cortez performed the experiments, analyzed the data, prepared figures and/or tables, and approved the final draft.

Carlos Pedraza-Lara performed the experiments, authored or reviewed drafts of the article, and approved the final draft.

Oliver Millan-Catalan performed the experiments, analyzed the data, prepared figures and/or tables, and approved the final draft.

Carlos Pérez-Plasencia analyzed the data, authored or reviewed drafts of the article, and approved the final draft.

The following information was supplied relating to ethical approvals (i.e., approving body and any reference numbers):

All procedures for the PMI rat model were evaluated and approved by the local ethic and scientific committee, as well as the committee for the care and use of laboratory animals (CICUAL) of the Faculty of Medicine from the National Autonomous University of Mexico (UNAM) with approval number 102-2018, and 027-CIC-201, respectively.

The following information was supplied regarding the deposition of microarray data:

The data is available at Gene Expression Omnibus: GSE208236.

The following information was supplied regarding data availability:

The sample signals of all miRNAs analyzed in control and 24 h of postmortem interval samples and the Ct values obtained from RT-PCR experiments are available in the Supplemental Files.

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
