# Peer review of "A miRNome analysis at the early postmortem interval"

_PeerJ, doi:10.7717/peerj.15409_

## Round 0.1 · original submission · Minor Revisions

RIN number and/or total RNA migration on agarose to verify the presence or not of autolyze should be included.

Reviewer 1 ·

Basic reporting

This article is well written, no need for language revision. Literature references are relevant for the topic and they include the latest research in PMI during the last years. The article structure is correct, including enough information in the introduction; well-detailed material and methods; a thorough description of the results with their corresponding figures; and the discussion compared the results with the latest research in PMI.

Experimental design

This is an interesting study addressing one of the main issues in forensic sciences, the accurate estimation of post-mortem interval (PMI). There are different approaches to assess this estimation, and the authors decided to test one of the latest trends, the analysis of miRNAs. The experimental design is adequate for the aim of the study. Based on their detailed explanation of the methods, it would be possible to reproduce the experiment by other researchers. The authors should explain maybe on the material and methods and/or the discussion why they chose only two times to study the miRNAs, 0 and 24 hours. Why don't they analyze intermediate times (like 12 hours) or beyond 24 hours (like 48, 72 hours)?

Validity of the findings

The results are sounded and fitted very well with the findings from previous research, particularly they confirmed the involvement of autophagy and cell cycle pathways on PMI estimation. These conclusions demonstrated the relevance of this research towards the aim of this study: improvement of PMI estimation.

Additional comments

No additional comments.

Reviewer 2 ·

Basic reporting

'no comment'

Experimental design

'no comment'

Validity of the findings

'no comment'

Additional comments

The current manuscript by Guardado-Estrada and collaborators demonstrates an important push to develop novel tools for the forensic field. The work addresses the participation of miRNAs at early (24hs) postmortem interval (PMI) that could be explored to identify potential biomarkers for PMI estimation. In general, the experimental approach is well-designed. Some minor comments should be considered.

1. Add a comment/section about potential limitations to the use of this technology (e.g. costly, available kits, specialized personnel for collection/analysis).

2. Have authors identified additional changes in the myomiRs family (e.g. miR-1 and miR-206, etc.)? And miR-223-3p, miR-24-3p (recently described by Koutalianos et al. 2021 PMID: 34703840).

3. Figure 1. (Volcano) identifies the dots (top 10) most upregulated and downregulated.

4. Figure 3. Identify the mRNA targets.

5. Figures need better resolution.

·

Basic reporting

English language need to be improved, some sentences need to be rewritten to make sens of the ideas, example: L53

Experimental design

Design could be improved by adding other rats at different time points (example: at 6h, 12h and 48h time points).
Also add RIN number and/or show total RNA migration on agarose to verify the presence or not of autolyze.
Microscopy photos will be interresting

Validity of the findings

The author should reorient their paper from ‘autolyse’ to ‘change in RNA expression in tissue’. They did not proved that the phenomena observed is due to RNA autolysis and I would be very surprised that it is autolyze. Instead their findings point to a change in cell expression that we observed in our lab (e.g. activation of immune cells post death...) c.f. Dachet et al., Selective time-dependent changes in activity and cell-specific gene expression in human postmortem brain,2021.
If the author want to prove the phenoma is due to autolyse they need to explain the presence of upregulated miRNAs and add RNA integrity (RIN) or another way to measure RNA autolysis in function of time (e.g. at minimum show the electrophoresis of total RNA for each sample and each time point). Like the muscle is a relatively complex tissues with various type of cells, author should also add microscopy figures of the samples involved in the RNA extraction using markers from major type of cells (red blood cell, muscle, immune cells...)

Additional comments

L18, L46: Just ‘found the body' is not enough to define the PMI, the definition of the PMI should be since death and until body is analyzed or sample preserved from post mortem effect. The PMI is increased during the time the body is discovered and the time the body is analyzed or sample preserved for future analysis.
L56, 58…: PMI is not synonymous to autolysis as there is no direct link between autolyze and time of death. The effect of PMI on transcriptom in the first 24h is probably related to change in immune system cell created by muscular hypoxia and regulation of various mechanism of gene expression due to homeostasis alteration. The cells are still alive for some times after death and the autolyze start majority 48h after death, c.f. multiple articles that did not found any significance change in RNA integrity during the first 48h post death indicating autolyse is not the primary factor leading to change in RNA after death.
L194-196: Hypoxia does not relate to autolytic process, but to stress process

L220: Interleukin secretion is a major event leading to regulation of immune cells and usually increase in the quantity of these cells in the tissue. You should discuss about immune system.

L232: You did not prove any degradation but your results indicate a change in cell composition (potentially activation of immune cells that would explain the up regulation of some population of miRNAs).

L308: what about the relation of SIRT1 and innate immune cells? The decrease of SIRT1 is pro-inflammatory

---

## Round 0.2 · accepted · Accept

The authors have addressed the reviewers' comments.